# Effects of time to chemoradiation on high-grade gliomas from the Buenos Aires Metropolitan Area

Diego M. Prost[1]*, Martín A. Merenzon[2], José I. Gómez-Escalante[3], Andrés Primavera[4], Mara Vargas Benítez[4], Andrés S. Gil[5], Pablo M. Marenco[6], María M. Califano[7], Carolina Moughty Cueto[2], Juan M. Zaloff Dakoff[2], Mario Colonna[2], Alejandro Mazzón[2], Roberto S. Zaninovich[8], Oscar R. De Cristófaro[1,9]

**1** Neuro-Oncology Unit, Medical Oncology Department, Instituto de Oncología Ángel H. Roffo, Universidad de Buenos Aires, Buenos Aires, Argentina, **2** Neuro-Oncology Unit, Surgical Oncology Department, Instituto de Oncología Ángel H. Roffo, Universidad de Buenos Aires, Buenos Aires, Argentina, **3** Pathology Department, Instituto de Oncología Ángel H. Roffo, Universidad de Buenos Aires, Buenos Aires, Argentina, **4** Medical Imaging Department, Instituto de Oncología Ángel H. Roffo, Universidad de Buenos Aires, Buenos Aires, Argentina, **5** Radiation Oncology, Centro privado de Radioterapia, Río Cuarto, Córdoba, Argentina, **6** Radiation Oncology Department, Instituto de Oncología Ángel H. Roffo, Universidad de Buenos Aires, Buenos Aires, Argentina, **7** Psico-Oncology Department, Instituto de Oncología Ángel H. Roffo, Universidad de Buenos Aires, Buenos Aires, Argentina, **8** Neurosurgery Department, Hospital de Clínicas José de San Martín, Universidad de Buenos Aires, Buenos Aires, Argentina, **9** Clinical Oncology Department, Instituto de Oncología Ángel H. Roffo, Universidad de Buenos Aires, Buenos Aires, Argentina

* dieprost@gmail.com

**Data Availability Statement:** All relevant data are within the paper.

**Funding:** The authors received no specific funding for this work.

## Abstract

High-Grade Gliomas (HGG) are the most frequent brain tumor in adults. The gold standard of clinical care recommends beginning chemoradiation within 6 weeks of surgery. Disparities in access to healthcare in Argentina are notorious, often leading to treatment delays. We conducted this retrospective study to evaluate if time to chemoradiation after surgery is correlated with progression-free survival (PFS). Our study included clinical cases with a histological diagnosis of Glioblastoma (GBM), Anaplastic Astrocytoma (AA) or High-Grade Glioma (HGG) in patients over 18 years of age from 2014 to 2020. We collected data on clinical presentation, type of resection, time to surgery, time to chemoradiation, location within the Buenos Aires Metropolitan Area (BAMA) and type of health insurance. We found 63 patients that fit our inclusion criteria, including 26 (41.3%) females and 37 (58.7%) males. Their median age was 54 years old (19–86). Maximal safe resection was achieved in 49.2% (n = 31) of the patients, incomplete resection in 34.9% (n = 22) and the other 15.9% (n = 10) received a biopsy, but no resection. The type of health care insurance was almost evenly divided, with 55.6% (n = 35) of the patients having public vs. 44.4% (n = 28) having private health insurance. Median time to chemoradiation after surgery was 8 (CI 6.68–9.9) weeks for the global population. When we ordered the patients PFS by time to chemoradiation we found that there was a statistically significant effect of time to chemoradiation on patient PFS. Patients had a PFS of 10 months (p = 0.014) (CI 6.89–13.10) when they received chemoradiation <5 weeks vs a PFS of 7 months (CI 4.93–9.06) when they received chemoradiation between 5 to 8 weeks and a PFS of 4 months (CI 3.76–4.26 HR 2.18 p = 0.006) when they received chemoradiation >8 weeks after surgery. Also, our univariate and

**Competing interests:** The authors have declared that no competing interests exist.

**Abbreviations:** HGG, high-grade gliomas; GBM, glioblastoma; AA, astrocytoma anaplastic; BAMA, Buenos Aires Metropolitan Area; PFS, period free survival; IDH1-2, isocitrate dehydrogenase type 1 and 2; MGMT, O6-Methylguanine-DNA methyl transferase.

multivariate analysis found that temporal lobe location (p = 0.03), GMB histology (p = 0.02) and biopsy as surgical intervention (p = 0.02) all had a statistically significant effect on patient PFS. Thus, time to chemoradiation is an important factor in patient PFS. Our data show that although an increase in HGG severity contributes to a decrease in patient PFS, there is also a large effect of time to chemoradiation. Our results suggest that we can improve patient PFS by making access to healthcare in Buenos Aires more equitable by reducing the average time to chemoradiation following tumor resection.

## Introduction

High-grade gliomas (HGG) are the most frequent malignant brain tumors in adults, and tumor recurrence after surgery is almost inevitable [1]. The gold-standard treatment involves a chemoradiation treatment strategy based on temozolomide, and has been used since the early 2000's. Its widespread use has duplicated the number of patients alive after 2 years after diagnosis [2–4]. In addition to chemoradiation, maximal safe surgical resection is one of the key factors in improving patient outcomes [5]. Also, starting adjuvant treatment within six weeks after surgery is strongly recommended [6, 7].

The state of Argentinian health care is linked to the several economic crises that the country has suffered in recent history [8]. The Buenos Aires Metropolitan Area (BAMA) has 14 million people that come from every socioeconomic background [9]. The healthcare that patients in the Buenos Aires Metropolitan Area receive depends on their socioeconomic status and where they live [10]. To date there is only one published study that analyzed brain cancer patients' access to treatment and their outcomes in Argentina [11]. This previous work showed significant differences between public and private institutions throughout the country and the lack of access to specialized brain tumor teams. In many cases after their initial diagnosis patients had to be referred to another facility for an MRI, to consult a neurosurgeon or to receive radiotherapy. However, this study did not assess the impact of access to healthcare on disease progression.

Here, we conducted a retrospective study with HGG patients from the BAMA treated in our center, as well as consultation cases. The primary aim of our study was to examine whether time to chemoradiation affected disease progression, specifically to see if there was an increased risk of progression per week of delay. We also evaluated clinical and histological characteristics which could affect progression-free survival.

## Methods

We selected patients who consulted our neuro-oncology unit at Instituto de Oncología Ángel Roffo with a histological diagnosis of HGG between January 2014 and March 2020. Eligible patients met the following inclusion criteria: > 18 years of age, ECOG performance status of 2 or less, living in the BAMA, histological diagnosis of glioblastoma (GBM), anaplastic astrocytoma (AA) or high-grade glioma NOS (HGGnos), admitted in ICU after surgery for no more than 72 hours, and no record of infection or perioperative complications. Molecular tests for IDH1-2 wild type (isocitrate dehydrogenase type 1 and 2), MGMT (O6-Methylguanine-DNA methyl transferase) non-methylated, but not 1p/19q co-deletion were included (Fig 1). From the original 93 patients only 63 were suitable for further analysis under these criteria. The most common reason for exclusion was spending more than 72 hours in the ICU due to infections or fistula after surgery that led to unavoidable delays in chemoradiation. We also had to exclude cases in which we were unable to measure the time from surgery to chemoradiation, symptom

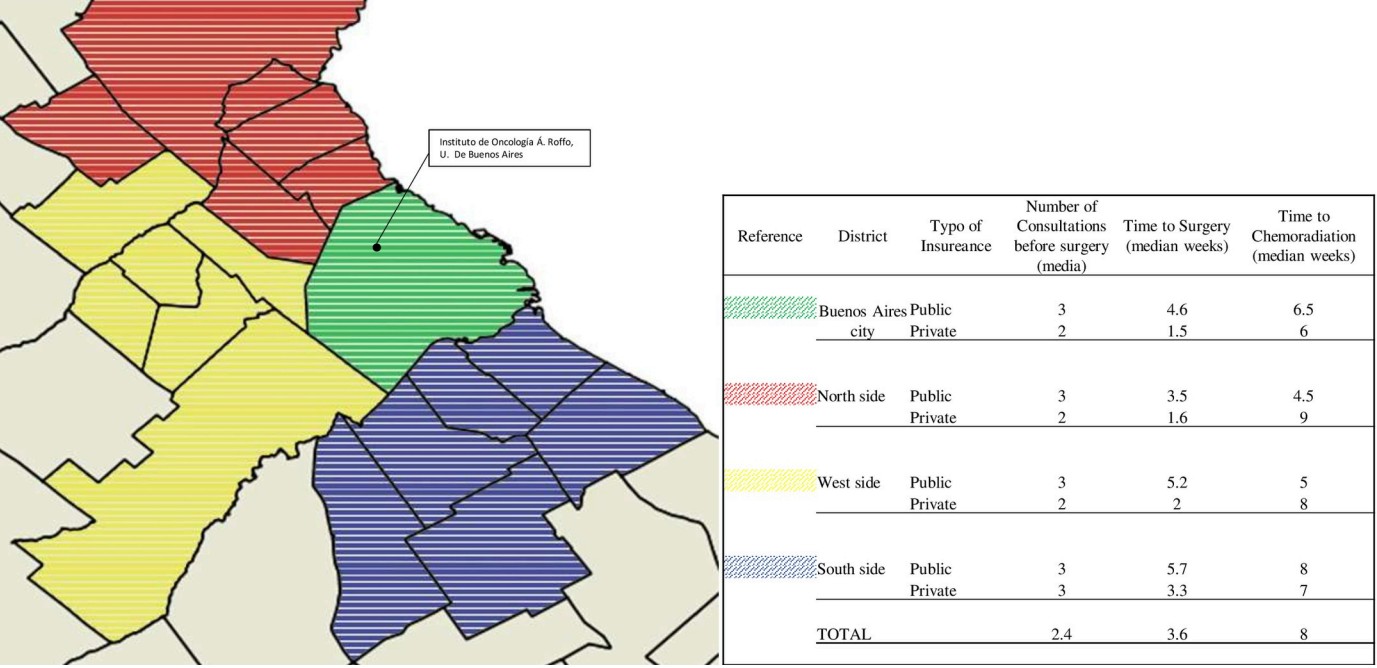

**Fig 1. Buenos Aires Metropolitan Area (BAMA) map.** Different colors show the districts where patients live and the location of Instituto de Oncología Á. Roffo in Buenos Aires City. Data presented: number of consultations with a specialist before surgery (median); time to surgery from the first symptom (median weeks), time to chemoradiation from surgery procedure (median weeks); distinguishing patients from the public and private sector.

development or both. The following data was collected from patient clinical records: sex, age at diagnosis, tumor lobe location, date of surgery, type of surgery (maximal safe resection for those with less than 10% of lesion remaining; incomplete resection for those with between 10–50% of lesion remaining, and biopsy when there was >50% of lesion remaining), area of residence in BAMA (City of Buenos Aires; North side; West side or South side), type of health insurance (public or private), number of consultations with a specialist before surgery (including visits to any referral consultation with a specialized care giver or procedure that could not be performed in the original place of treatment), time to surgery after the first symptom, time to chemoradiation (stratified in <5 weeks; 5–8 weeks and >8 weeks from surgery to chemoradiation to analyze the possible risk increase per week of delay) and progression-free survival (PFS, defined as the time from surgery to the development of new lesions as detected by MRI). Progressive disease was defined using the RANO criteria [12]. The data we collected was anonymized and codified to preserve participants' identity according to the Argentine law [13]. We received approval from the Instituto Roffo Clinical ethics committee, which decided that due to the retrospective nature of this work no informed consent was required. However, every patient recruited was informed and gave oral or written approval for the use of their clinical data after anonymization.

The statistical analysis of time to progression was performed with the Kaplan-Meier method. We used the Cox proportional hazards model to calculate adjusted univariate hazard ratios (HR) and their 95% Cis. Statistical analyses were conducted using IBM SPSS Statistics 25 (IBM, Armonk, New York, USA). P-values <0.05 were considered statistically significant.

## Results

A total of 63 HGG cases were included in this study, including 27 females (41.3%) and 36 males (58.7%) with a median age of 54 (ranging from 19 to 86) years old. Most tumors were

located in the parietal lobe 47.6% (n = 30), and 49.2% (n = 31) achieved a maximal safe resection. The most frequent histological diagnosis was GBM (n = 54; 85.7%), followed by AA (n = 7; 11.1%) and HGGnos (n = 2; 3.1%). The number of patients with public and private health insurance was similar, with 55.6% (n = 35) of patients having public versus 44.4% (n = 28) having private health insurance (further information can be found in Table 1).

After symptom onset our patients had a median of 3.6 consultations with specialists. Patients with public heathcare had a median of 3 consultations versus 2 for patients with private healthcare in almost all of the districts in the BAMA. The difference in time to surgery was 5.03 (CI 4.03–5.75) weeks for patients with public healthcare versus 1.86 (CI 1.5–2.22) weeks for patients with private healthcare. The time to chemoradiation was binned into three groups: 36.5% (n = 23) received chemoradiation in <5 weeks; 28.6% (n = 18) in 5–8 weeks; and 34.9% (n = 18) in >8 weeks.

The univariate analysis for PFS showed a significant correlation between temporal lobe location (p = 0.033) and time to chemoradiation >8 weeks after surgery (p = 0.006). Also, the

**Table 1. Patient's characteristics.**

| Variables | n | % |
|---|---|---|
| *Sex* | | |
| Female | 26 | 41.3 |
| Male | 37 | 58.7 |
| *Age median (range)* | 54 | (19–86) |
| *Tumor Location (lobe)* | | |
| Frontal | 10 | 15.9 |
| Parietal | 30 | 47.6 |
| Temporal | 19 | 30.2 |
| Occipital | 4 | 6.3 |
| *Number of Consultations before surgery (median)* | 3.6 | |
| *Surgery* | | |
| Maximal safe resection | 31 | 49.2 |
| incomplete resection | 22 | 34.9 |
| Biopsy | 10 | 15.9 |
| *Healthcare Insurance* | | |
| Public | 35 | 55.6 |
| Private | 28 | 44.4 |
| *District* | | |
| Buenos Aires City | 23 | 36.5 |
| North Side | 9 | 14.3 |
| West Side | 16 | 25.4 |
| South Side | 15 | 23.8 |
| *Molecular Analysis* | | |
| IDH wt | 38 | 60.3 |
| unknown IDH status | 25 | 39.6 |
| MGMT non-methylated | 8 | 12.6 |
| *Time to chemoradiation* | | |
| <5 weeks | 23 | 36.5 |
| 5–8 weeks | 18 | 28.6 |
| >8 weeks | 22 | 34.9 |

The Table shows number of cases were MGMT methylation were performed and was non-methylated.

surgical outcome was significant for patients that received a biopsy (p = 0.002, HR 0.276 CI 0.134–0.566) and maximal safe resection (p = 0.021, HR 1.81 CI 1.09–3). Histological diagnosis of GBM (p = 0.002, HR 0.299 CI 0.125–0.715) and AA/HGGnos (p = 0.002, HR 3.34 CI1.39–8.01) were significantly associated with PFS. AA and HGGnos were analyzed together due to the low number of patients with HGGnos. IDH1-2 status did not represent a considerable bio-marker for our population (p = 0.529). Multivariate analysis for PFS showed statistical significance for histological diagnosis, lobe location and biopsy intervention. See Table 2.

Kaplan-Meier analysis revealed a median PFS of 7 months (CI 5.72–8.27). Sub-group analy-sis showed a tendency of median PFS of 10 months (p = 0.058, CI 4.66–15.33) for patients with tumors located in the temporal lobe compared to 5 months (CI 3.21–5.68) for those with tumors located in the parietal lobe (Fig 2). GBM patients had a median PFS of 6 months (p = 0.003, CI 4.80–7.19) vs AA/HGGnos patients who had a median PFS of 13 months (CI 8.31–17.68). IDH1-2 status did not show significant differences between patients with wild type versus unknown status, with a median PFS of 6 (CI 3.98–8.01) and 7 months (CI 5.37–8.62, p = 0.529), respectively. Patients who achieved maximal safe resection had a PFS of 8 months (p = 0.001, CI 5.81–10.18); compared to 5 months (CI 3.62–6.37) for those who received an incomplete resection, and 3 months (CI 1.45–4.55) with patients that only received a biopsy. Time to chemoradiation also demonstrates a significant difference for patients who started treatment <5 weeks from surgery with a median PFS of 10 months (p = 0.014, CI 6.89–13.10); compared to 7 months (CI 4.93–9.06) for patients who began treatment after 5 to 8

**Table 2. Univariate and multivariate analysis for PFS.**

| Variable | Univariate | | | | Multivariate | | | |
|---|---|---|---|---|---|---|---|---|
| | *P* | HR | CI 95% | | *P* | HR | CI 95% | |
| *Sex* | .843 | .949 | 0.75 | 1.26 | | | | |
| *Lobe location* | | | | | | | | |
| Frontal | .344 | .716 | .716 | 1.42 | | | | |
| Parietal | .111 | .661 | .397 | 1.09 | | | | |
| Temporal | **.033** | **.603** | **.011** | **.679** | **.003** | **.598** | **0,427** | **0,839** |
| Occipital | .730 | 1.19 | .429 | 3.34 | | | | |
| *Number of consultations to specialist before surgery* | .232 | .843 | .637 | 1.11 | | | | |
| *Surgical Resection* | | | | | | | | |
| Maximal Safe Resection | **.021** | **1.81** | **1.09** | **3.00** | .29 | 1.36 | .768 | 2.41 |
| Incomplete Resection | .541 | .891 | .503 | 1.43 | | | | |
| Biopsy | **.002** | **.276** | **.134** | **.566** | **.0004** | **.306** | **.135** | **.691** |
| *Histology* | | | | | | | | |
| GBM | **.002** | **.299** | **.125** | **.715** | **.003** | **.260** | **.106** | **.638** |
| AA/HGGnos | **.002** | **3.34** | **1.39** | **8.01** | **.007** | **3.34** | **1.40** | **8.0** |
| *Time to Surgery* | | | | | | | | |
| <4 weeks | .464 | 1.21 | .727 | 2.02 | | | | |
| >4 weeks | .436 | .825 | .494 | 1.37 | | | | |
| *Time to Chemoradiation* | | | | | | | | |
| <5 weeks | .085 | 1.57 | .933 | 2.64 | | | | |
| 5–8 week | .500 | 1.21 | .689 | 2.12 | | | | |
| >8 weeks | **.006** | **2.18** | **1.27** | **3.75** | .115 | 1.52 | .852 | 2.72 |
| *IDH1 wt/unknow* | .529 | .849 | .508 | 1.41 | | | | |
| *Health Care Insurance (public/private)* | .842 | .950 | .572 | 1.57 | | | | |

Tumor histology: Glioblastoma (GBM), Anaplastic Astrocytoma (AA) and High-Grade Glioma Nos (HGGnos). Statistically significant results in bold.

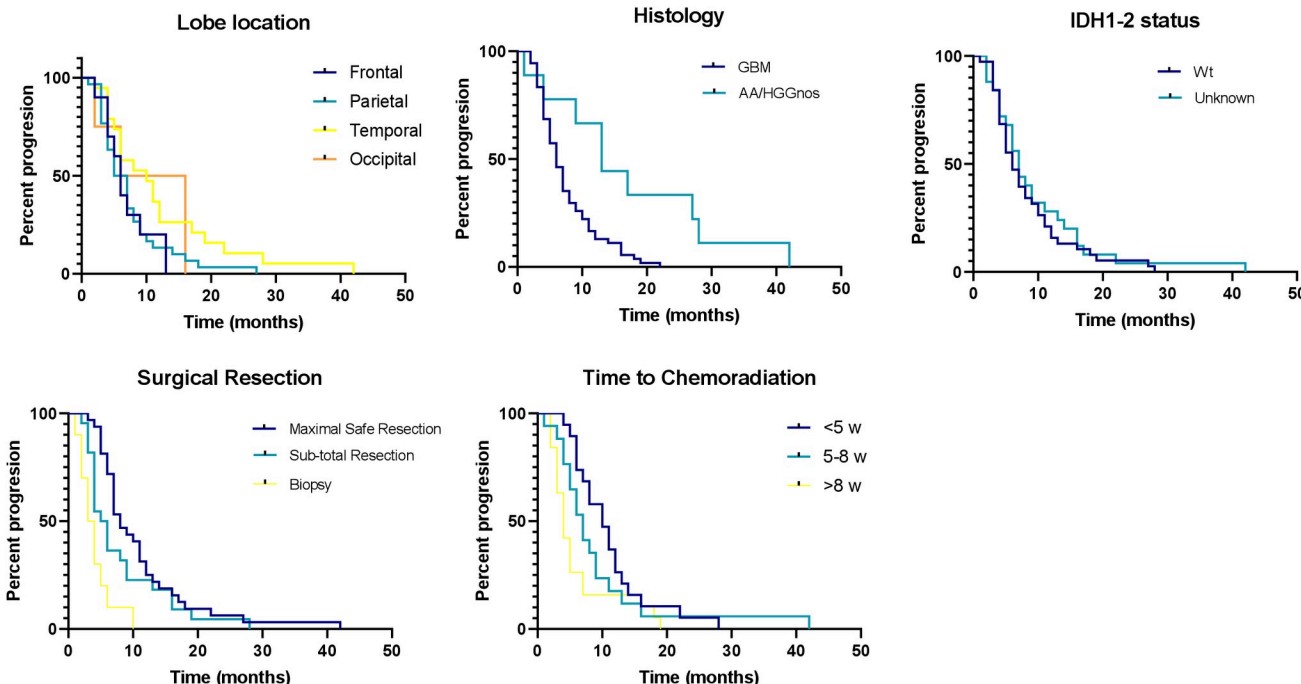

**Fig 2.** Kaplan-Meier curves for PFS for (A) tumor lobe location, (B) histology, (C) IDH status, (D) surgical procedure and (E) time to chemoradiation.

weeks, and 4 months (CI 3.76–4.26) for patients who waited >8 weeks to begin treatment. Finally, there was no difference in PFS for patients with public vs. private health insurance (7 vs 6 months respectively, p = 0.842; CI 0.57–1.57).

## Discussion

Access to healthcare access is an important issue for any society, but even more so in the Buenos Aires metropolitan area considering the disparities in education and infrastructure [14, 15]. Our study reveals the differences in some aspects of care given to patients with HGG in BAMA. One of the major factors in progression-free survival is maximal safe resection. Approximately half of the patients (both with private and public health insurance) in our study received maximal safe resection, and as was expected had a better PFS. On the other hand, a minority of patients (n = 10) received a biopsy as their only surgical intervention, and their prognosis was considerably worse than that of all other patients. It is not unusual to see patients from the BAMA for a consultation who only received a diagnostic biopsy for HGG. It is already known that incomplete resection might be more beneficial than a simple biopsy, especially to avoid intracranial hypertension symptoms [16].

We used the number of consultations with a specialist before surgery as a variable because not many centers in the Buenos Aires metropolitan area are able to completely diagnose and treat patients with HGG. Typically, patients from the public sector consult in their local area hospital after their first symptom, and often must be referred to another center for an MRI or to receive an evaluation from a surgeon. Therefore surgical resection hardly ever takes place before 3 weeks from the appearance of symptoms. Setting up an appointment with an oncologist could take another 3 weeks. The situation is completely different for patients from the private sector, who are typically diagnosed and operated on at the same center within the first week after symptom onset. In spite of receiving surgery more quickly, these patients

nonetheless find it hard to achieve the 6-week benchmark for chemoradiation post-surgery. We did not find any correlation between the number of consultations and PFS in our patients. However, this is clearly a factor that could be improved to enhance treatment efficiency.

Time to chemoradiation was similar for both public and private patients from almost all districts. We believe that this can be explained by two major reasons. First, patients from the private sector are commonly referred to an oncologist after their surgical scar is completely healed. This takes around twenty days, after which the specialist involved begins planning radiotherapy treatment, making it difficult to achieve the six-week or less benchmark despite undergoing surgery shortly after symptom onset. All patients from the public sector who started early chemoradiation treatment were those attended in our center or referred to us after surgery, and who were previously discussed in our tumor board. This could only happen after the establishment of referral networks among hospitals so that low-income patients can also access the standard of care. Due to the lack of previous statistics, we can only guess how the treatment of these patients has improved. But it should be noted that none of the patients included in this study who came from the public sector and were treated in other institutions started chemoradiation before 8 weeks. For this group, their biggest challenge was to find an oncologist with experience in treating gliomas.

Secondly, there is a serious lack of centers capable of completely diagnosing gliomas in BAMA. Molecular diagnosis, including IDH1-2 mutations (among others), is only available in academic centers and is usually limited by cost. Due to this many patients experience delays in beginning their chemoradiation treatment. Patients from every district and socioeconomic level were affected, making the histological diagnosis another aspect that if sped up will improve the overall performance of the healthcare system.

Due to the frequency and aggressiveness of these tumors, HGG must be one of the main diagnoses when facing a patient with a primary brain lesion. Thus it is imperative to act fast and contact the entire specialist team. Delaying treatment to wait for the molecular analysis (for instance IDH1-2 mutations results) could be detrimental for the patient, especially when the prognostic role of IDH1-2 status in HGG is debated, as described in recent studies which propose CDKN2A/B for anaplastic astrocytoma [17–19].

We recognize the limitations of our work due to its retrospective nature and small sample size. Another limitation is our lack of molecular diagnoses for HGG, especially MGMT methylation. The exclusion of patients with a methylated test attempted to avoid this source of bias as they were all diagnosed and treated at the same private institution. The clinical features that we found to statistically correlate with patient PFS have been widely described in previous studies [20, 21] with similar results. Choosing PFS instead of other parameters such as overall survival was due to a lack in follow up data for patients not treated in our center. Also, given the study design, we only found a significantly lower PFS in the >8-week group, as was already known. However, we believe that the real value of our work lies in identifying the first obstacles to be solved in order to improve access to healthcare.

We will continue to recommend the development of easy ways to refer HGG patients in the Buenos Aires Metropolitan Area. At the same time, we strongly recommend an interdisciplinary approach even prior to histological diagnosis, with the aim of delivering the best treatment to our patients and reducing socioeconomic gaps and excessive bureaucracy.

## Conclusions

Finally, we conclude that biopsy as the only surgical procedure and chemoradiation later than 8 weeks after surgery were two negative factors for HGG in our population. These negative factors were both previously described. Many failings lead to delays in chemoradiation treatment

in both the public and private sectors, and a lack of communication between multidisciplinary teams seems to be the major issue.

## Acknowledgments

We would like to thank Elijah D. Lowenstein for critically reading the manuscript. Also, thanks to Ana Eijan, Denise Belgorsky, Marcela Villaverde and Elsa Hincapié from the Roffo Institute Research Department, and Mauricio Paez and Mayra Aldecoa for collecting and evaluating patients at the Neurology Department of Ramos Mejía Hospital.

## Author Contributions

**Conceptualization:** Diego M. Prost.

**Data curation:** Diego M. Prost, Martín A. Merenzon, José I. Gómez-Escalante, Andrés Primavera, Mara Vargas Benítez, Pablo M. Marenco, María M. Califano, Carolina Moughty Cueto, Juan M. Zaloff Dakoff, Mario Colonna, Alejandro Mazzón, Roberto S. Zaninovich.

**Formal analysis:** Diego M. Prost, María M. Califano.

**Investigation:** Diego M. Prost, Andrés S. Gil, Pablo M. Marenco, María M. Califano.

**Methodology:** Diego M. Prost, María M. Califano.

**Project administration:** Diego M. Prost.

**Resources:** Martín A. Merenzon, Andrés S. Gil, Roberto S. Zaninovich.

**Software:** Andrés Primavera, Mara Vargas Benítez, Andrés S. Gil.

**Supervision:** María M. Califano, Roberto S. Zaninovich, Oscar R. De Cristófaro.

**Writing – original draft:** Diego M. Prost.

**Writing – review & editing:** Diego M. Prost, Oscar R. De Cristófaro.

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
