## [Decision Letter · Decision Letter 0]

29 Dec 2020

PONE-D-20-37225

Effects of time to chemoradiation on high-grade gliomas from Buenos Aires Metropolitan Area

PLOS ONE

Dear Dr. Prost,

Thank you for submitting your manuscript to PLOS ONE. After careful consideration, we feel that it has merit but does not fully meet PLOS ONE’s publication criteria as it currently stands. Therefore, we invite you to submit a revised version of the manuscript that addresses the points raised during the review process.

Please change the manuscript according to the suggestions brought forward by the reviewers. If this is not possible, please explain the reason in detail.

We look forward to receiving your revised manuscript.

Kind regards,

Michael C Burger, M.D.

Academic Editor

PLOS ONE

Journal Requirements:

2. In your ethics statement in the manuscript and in the online submission form, please ensure that you have discussed whether all data/samples were fully anonymized before you accessed them and/or whether the IRB or ethics committee waived the requirement for informed consent. If patients provided informed written consent to have data/samples from their medical records used in research, please include this information.

3. In the ethics statement in the manuscript and in the online submission form, please provide additional information about the patient records/samples used in your retrospective study, including the date range (month and year) during which patients' medical records/samples were accessed.

4. Please ensure that you refer to Figure 3 in your text as, if accepted, production will need this reference to link the reader to the figure.

5. Please include a copy of Table 1 and 2 which you refer to in your text on page 5.

Reviewers' comments:

Reviewer's Responses to Questions

**Comments to the Author**

1. Is the manuscript technically sound, and do the data support the conclusions?

Reviewer #1: Yes

Reviewer #2: Yes

2. Has the statistical analysis been performed appropriately and rigorously? 

Reviewer #1: Yes

Reviewer #2: Yes

3. Have the authors made all data underlying the findings in their manuscript fully available?

Reviewer #1: Yes

Reviewer #2: Yes

4. Is the manuscript presented in an intelligible fashion and written in standard English?

Reviewer #1: Yes

Reviewer #2: Yes

5. Review Comments to the Author

Reviewer #1: The authors report their retrospective experience with high grade glioma in metropolitan Buenos Aires. in a series of 63 patients accumulated over six years, they note that among this cohort that a time greater than 8 weeks to the start of chemoirradiation from surgery as well as biopsy were two strong negative predictors of outcome. Both of these were more significant than whether patients were treated in public vs. private sectors.

There are several deficiencies in this analysis. First, the number of patients analyzed were accrued at a rate of only ten per year. It's hard to believe that this was a substantial fraction of the patients operated on at that time. No explanation is given for this and the low number of patients certainly opens up questions as to the robustness and generalizability of their results.

It's unclear why they excluded MGMT positive patients. While these patients may do better than ones that are unmethylated, it still would be useful to assess impact of delay and biopsy.

Time is not really a binary value, so it's unclear why they didn't assess delay as a continuous variable.

The lack of clinical data and retrospective nature of study is prone to bias. For instance, it could be that the time to RT resulted from prolonged Rehab hospitalizations due to physical deficits that by themselves would be negative prognostic factors.

The conclusion should be reworded. I’m sure they don’t mean to say that those are the only two bad prognostic factors for GBM.

While only one study may have reported the effects of wait time on outcome in Argentina, there are several in the literature, all favoring initiating treatment prior to 4-6 weeks. Likewise, there have been many studies that have pointed to the negative prognostic impact of biopsy.

Reviewer #2: The authors have provided an interesting description of the time from diagnosis to chemoirradiation from the Buenos Aires metropolitan area. Though it is already known that time to chemoirradiation significantly influences survival this study is important in order to develop better working protocols. To do so, one needs to understand the current situation, and this is achieved by this paper. Sadly, the authors show that only a minority of patients receives chemoirradiation at the proper time. Thus, the authors have much work to do to improve these times in the Buenos Aires Cosmopolitan Area, now that they provide an accurate diagnosis.

As the authors present a detailed analysis of the current situation, I would like to encourage them that, at the end of their manuscript they include a proposal on how to remedy the current defective situation. It will be important to read in a future report in a few years, how the time of chemoirradiation has been reduced for these patients.

This manuscript will benefit from a scientific english editor of the main manuscript's text.

6. PLOS authors have the option to publish the peer review history of their article (what does this mean?). If published, this will include your full peer review and any attached files.

Reviewer #1: No

Reviewer #2: **Yes: **Pedro R. Lowenstein, MD, PhD

---

## [Author Response · Author response to Decision Letter 0]

16 Mar 2021

Dear Dr. Michael C Burger, 

Academic Editor, 

PLOS ONE 

Thank you for giving me the opportunity to submit a revised draft of my manuscript titled “Effects of time to chemoradiation on high-grade gliomas from Buenos Aires Metropolitan Area” to PLOS ONE. I appreciate the time and effort that you and the reviewers have dedicated to providing your valuable feedback on my manuscript. I am grateful to the reviewers for their insightful comments on my paper. Here in Argentina, our medical training offers almost nothing on academic writing and because of this I have been able to incorporate changes to reflect most of the suggestions provided by the reviewers. I have highlighted the changes within the manuscript, as it was requested. Here is a point-by-point response to the reviewers’ comments and concerns. 

Comments from Reviewer 1 

Comment 1: First, the number of patients analyzed were accrued at a rate of only ten per year. It's hard to believe that this was a substantial fraction of the patients operated on at that time. No explanation is given for this and the low number of patients certainly opens up questions as to the robustness and generalizability of their results. 

Response: We decided to expand information on the patients flow in our manuscript. Also, it is necessary to explain that there is no register of High-Grade Glioma in adult patients in Argentina. An interpretation of what is a representative population sample for us is hard to determine as most often statistics for HGG referred to locally are from the US or Europe and cannot be considered to illustrate our situation. Although our study is small, it at least tries to approach the current obstacles patients currently face in our city. 

Comment 2: It's unclear why they excluded MGMT positive patients. While these patients may do better than ones that are unmethylated, it still would be useful to assess impact of delay and biopsy.

Response: The reason for excluding MGMT methylated patients was because all of them came from the same private institution able to perform the molecular test. We have only been able to perform this molecular analysis in Insituto Roffo since late 2019. Patients with methylated results have clearly longer PFS on the pre-test review and this could be a bias in our analysis. 

Comment 3: Time is not really a binary value, so it's unclear why they didn't assess delay as a continuous variable. 

Response: We decided to analyze in three different categories of time according to weeks from surgery with the intention of demonstrating the risk increase, but it was not mentioned before recognizing the problem of over generalization with such a small population. Briefly, we described the intention in the highlighted modifications. 

Comment 4: The lack of clinical data and retrospective nature of study is prone to bias. For instance, it could be that the time to RT resulted from prolonged Rehab hospitalizations due to physical deficits that by themselves would be negative prognostic factors.

Response: We recognize our own clinical limitations for patients who have been included and treated in other centers as well those limitations inherent to a retrospective design. It is for this reason that we set up a strict criterion for the selection. For example, no patient was included with ECOG score above 2 or more than 72hr after surgery to avoid any element of misjudging caused by an infection or fistula. We recognize it was not clear enough in the original manuscript, so we explain it on the method section.

Comment 5: The conclusion should be reworded. I’m sure they don’t mean to say that those are the only two bad prognostic factors for GBM.

Response: We decided to rewrite the conclusion. 

Comment 6: While only one study may have reported the effects of wait time on outcome in Argentina, there are several in the literature, all favoring initiating treatment prior to 4-6 weeks. Likewise, there have been many studies that have pointed to the negative prognostic impact of biopsy.

Response: We agree with this comment, and those are both clinical features widely published and accepted for the HGG clinical management. We would like to emphasize the fact that our objectives were to evaluate how patients from our larger city access to this and what this impact was like. For example, almost every patient from the private sector received their surgical intervention within the week of symptoms debut. However, those patients achieved poorly the 6-week benchmark for chemoradiation. This is of our interest because it is something to be improved urgently. 

Comments from Reviewer 2 

Comment 3: This manuscript will benefit from a scientific english editor of the main manuscript's text.

Response: For this revision we ask for assistance from a native English speaker with scientific writing experience. 

I would like to thank you all for this opportunity and the time to read our manuscript. Your views and opinions were received as wise and encouraging for us. 

Kind regards

Diego M. Prost

---

## [Editor Report · Decision Letter 1]

19 Mar 2021

Effects of time to chemoradiation on high-grade gliomas from the Buenos Aires Metropolitan Area

PONE-D-20-37225R1

Dear Dr. Prost,

We’re pleased to inform you that your manuscript has been judged scientifically suitable for publication and will be formally accepted for publication once it meets all outstanding technical requirements.

Kind regards,

Michael C Burger, M.D.

Academic Editor

PLOS ONE
---

## [Editor Report · Acceptance letter]

25 Mar 2021

PONE-D-20-37225R1 

Effects of time to chemoradiation on high-grade gliomas from the Buenos Aires Metropolitan Area. 

Dear Dr. Prost:

I'm pleased to inform you that your manuscript has been deemed suitable for publication in PLOS ONE. Congratulations! Your manuscript is now with our production department. 

Kind regards, 

on behalf of

Dr. Michael C Burger 

Academic Editor

PLOS ONE